# The Management of Children with Cancer during the COVID-19 Pandemic: A Rapid Review

**DOI:** 10.3390/jcm9113756

**Published:** 2020-11-21

**Authors:** Matteo Amicucci, Angela Mastronuzzi, Italo Ciaralli, Federico Piccioni, Andreea Cristina Schiopu, Emanuela Tiozzo, Orsola Gawronski, Valentina Biagioli, Immacolata Dall’Oglio

**Affiliations:** 1Department of Onco Haematology and Cell and Gene Therapy, Bambino Gesù Children’s Hospital, IRCCS, 00165 Rome, Italy; angela.mastronuzzi@opbg.net (A.M.); italo.ciaralli@opbg.net (I.C.); federico.piccioni@opbg.net (F.P.); 2Department of Paediatric Emergency, Bambino Gesù Children’s Hospital, IRCCS, 00165 Rome, Italy; acristina.schiopu@opbg.net; 3Professional Development, Continuing Education and Nursing Research Service, Bambino Gesù Children’s Hospital, IRCCS, 00165 Rome, Italy; emanuela.tiozzo@opbg.net (E.T.); orsola.gawronski@opbg.net (O.G.); valentina.biagioli@opbg.net (V.B.); immacolata.dalloglio@opbg.net (I.D.)

**Keywords:** management, cancer, COVID-19, pediatric

## Abstract

Despite the fact that cancer patients seem to be at a higher risk of being infected with SARS-CoV-2, limited data are available in the pediatric oncology setting. A systematic rapid review was conducted to analyze scientific literature regarding the management, interventions, and strategies adopted to prevent the spread of COVID-19 in the pediatric cancer population. Our search on PubMed, Scopus, Cochrane, and EMBASE databases yielded 505 articles. After removing duplicates, 21 articles were included. Articles focused on infection prevention (*n* = 19; 90.5%), management (*n* = 18; 85.7%), overall management of specific treatments for cancer (*n* = 13; 61.9%), and education (*n* = 7; 33.3%). The interventions adopted to prevent the spread of COVID-19 were similar across organizations and in line with general recommendations. Most of them reported interventions that could be used as valid strategies for similar emergencies. The strategies included limiting the risk of contagion by restricting access to the wards and implementing hygiene measures, the identification of separate pathways for the management of patients suspected or confirmed to be infected with COVID-19, the postponement of people accessing the hospital for non-urgent or unnecessary tests or medical examinations, and the preventive screening of patients before chemotherapy treatment or transplantation of hematopoietic stem cells. It is necessary to identify key indicators in order to better evaluate the effectiveness of the interventions implemented over time. A summary of the recommendations is provided.

## 1. Introduction

Patients affected by cancer diseases are particularly at risk of SARS-CoV-2 infection due to higher morbidity and mortality associated with respiratory virus infections, for which the risk of hospitalization of cancer patients is four times higher compared to subjects of the same age [1,2]. Although information about children with cancer is scarce, data confirm a lower incidence of COVID-19 in children living with cancer [3,4,5,6].

The management and precaution measures to avoid being contaminated are not clearly defined and are based exclusively on limited experiences [7,8]. A recent survey in 25 countries, involving around 10,000 patients at risk, confirmed that a total of nine children with cancer were infected [9]. The authors conclude that this category of patients, due to their immunosuppression, remain at a high potential risk of COVID-19 [9]. Therefore, hospitals must constantly ensure that safe cancer care is provided [10]. However, an important aspect to consider is that most of the cancer patients cannot suspend their treatment and miss the window of treatment for the fear of contracting COVID-19 [11]. Various pediatric cancer organizations have published some recommendations regarding COVID-19, but none of these provide specific evidence for the interventions they describe [12,13,14,15,16,17,18,19,20]. Consequently, the problem related to COVID-19 is the need to have a clear understanding of what the additional effective and sustainable preventive precautions to be adopted are due to the increased risk of infection in children with cancer.

The purpose of this rapid review was to collect and analyze scientific literature regarding the management, interventions, and strategies to prevent the spread of COVID-19 in children with cancer.

## 2. Materials and Methods

### 2.1. Design

The present rapid review was conducted according to the Preferred Reporting Items for Systematic Reviews and Meta-Analyses (PRISMA) guidelines [21]. Rapid review has been best designed following the steps described by Khangura et al. (2012) and Cochrane Rapid Review Methods—Interim Recommendations (2020) [22,23].

### 2.2. Inclusion and Exclusion Criteria

In order to be included in the review, articles had to focus on (1) COVID-19 and children with cancer (hematological or oncological), either current, in remission, under chemotherapy, or with immunodeficiency; (2) interventions implemented to prevent and manage SARS-CoV-2 infections; and, if available (3) COVID-19 rates or pediatric outcomes (e.g., health, morbidity, or mortality), or other outcomes related to the implemented intervention. The exclusion criteria were (1) other pediatric patients affected by other diseases presenting immunodepression or (2) articles regarding the prevention and management of infections other than COVID-19.

### 2.3. Data Collection

The review questions were formulated through the P.I.C.O. method (see Appendix A). The following electronic databases were searched: PubMed, Scopus, Cochrane, and EMBASE. The rapid review was conducted on 15 June 2020 and updated on 26 July 2020. All the papers published from January 2019 onwards with titles and abstracts in English were reviewed.

### 2.4. Data Analysis

For the present rapid review, we adopted several methodological shortcuts: grey literature was excluded, and the results of the included articles were analyzed by only one author, while a second author verified the plausibility and correctness of the first author’s analysis. Due to the heterogeneity of the measures and analytical methodologies of the articles included in this review, we did not conduct a meta-analysis and narrative synthesis to avoid reaching biased results [21,24]. At the end of the data collection process, the following information was summarized from each paper: first author surnames, year of publication, region/country of the article, aim, article design or article type, population, setting, and interventions (Table 1). The level of evidence was calculated according to the hierarchies of evidence provided by the Oxford Centre for Evidence-Based Medicine Levels of Evidence (OECBM), which helps to identify bias related to article design [25].

## 3. Results

### 3.1. Articles Characteristics

Overall, 505 potentially eligible articles were retrieved from the electronic databases: 188 from PubMed, 144 from Scopus, 170 from EMBASE, and 3 from the Cochrane Library. After we removed duplicates and applied the exclusion criteria, 21 articles were included in the rapid review (see Appendix A). The main purpose of all the included articles was to investigate, describe, and promote strategies for the management and treatment to prevent the spread of COVID-19 in children with cancer. The included articles were conducted in five continents: Europe (*n* = 11; 52.4%), Asia (*n* = 10; 47.6%), America (*n* = 2; 9.5%), Oceania (*n* = 2; 9.5%), and Africa (*n* = 1; 4.8%), and represent a good geographical representation of the problem.

All the articles were descriptive, and the level of evidence was low (i.e., OECBM level 5). In particular, most of the included papers were letters to the editor (*n* = 5; 23.8%) [26,27,28,29,30] or recommendations (guidelines or expert consensus) (*n* = 5; 23.8%) [31,32,33,34,35]. The other papers were three correspondences [6,36,37], two reviews [38,39], two surveys [9,40], two commentaries [41,42], and two editorials [43,44].

The sample in all the included articles were children with cancer (hematological, oncological malignancies, or both). Seven articles (33.3%) involved parents and/or carers and staff. In all the included articles, the setting was a hospital unit. Seven articles also included an outpatient unit and home care (Table 1). None of the articles included a description of the criteria used to evaluate the efficacy of the interventions they proposed.

### 3.2. Interventions, Screening, Body Temperature, Swabs, and Additional Tests for the Diagnosis

The interventions mainly focused on infection prevention (*n* = 19; 90.5%), management measures (*n* = 18; 85.7%), management of onco-hematological treatments (*n* = 13; 61.9%), and education (*n* = 7; 33.3%) in children with cancer during the COVID-19 emergency (see Appendix A).

Initial screening to collect information about individuals’ health conditions is strongly recommended. In some cases, this is done by phone before an individual arrives at the hospital (e.g., in the case of outpatient care) [38]; in other cases, this is done in the outpatient clinic before hospital admission [9,26,28,29,31,37,39,43]. Screening often includes information regarding whether a person has travelled to a risk area before coming to the hospital, or if a person has been in contact with someone who is a probable or a confirmed COVID-19 case [38]. Often, screening is extended to all family members [26,29]. This is almost always followed by the measurement of body temperature and the detection of any of the typical COVID-19 symptoms, such as cough or respiratory symptoms [9,31,38,39], in order to identify suspected cases of COVID-19.

Once the screening has been completed, if the child needs to be hospitalized or invasive procedures have to be performed, the COVID-19 nasal swab is recommended. In some cases, the swab has to be performed also on parents [37]. The availability of negative test results is mandatory in order to carry out any activity in the hospital. In cases where swabs are in short supply or specific guidelines do not allow the use of swabs for everyone and in cases in low- and middle-income countries, social distancing, the use of masks, and hand hygiene, accompanied by information screening, remain strongly recommended [36].

### 3.3. Personal Protective Equipment (PPE), Handwashing Practice, and Isolation

To reduce the spread of SARS-CoV-2 infections, hygiene measures are recommended, such as frequent meticulous handwashing; not touching one’s eyes, mouth, and nose; and the use of PPE, especially wearing a mask. All the included articles refer to the preventive measures and use of PPE recommended by the World Health Organization (WHO) as a general preventive measure in the main settings where health workers may enter into contact with patients suspected or affected by COVID-19. These recommendations include the use of masks with filtering facepieces (FFP2 and FFP3) in every situation of risk (i.e., aerosol-generating procedures). It seems that these two preventive measures adopted for children with cancer are the same as those adopted for the rest of the population [6,9,26,29,31,33,37,39,41,43]. Instead, PPE kits should be sufficiently available [38]. In addition, PPE should be provided together with written instructions that explain how to wear them and which ones to use [33,37,42].

Any patient with a sore throat, cough, persistent malaise, and any changes in smell or taste should be considered a suspected case of COVID-19 and undergo isolation measures for droplet/contact infections [39]. The isolation of pediatric patients and their parents in cases of suspected COVID-19 contagion has to be implemented by promoting (if possible) the use of single rooms [35,36,43,44]. If COVID-19 positivity is confirmed, patients have to be transferred to hospital units specifically dedicated to COVID-19 patients [31,39]. In addition, if the health conditions of children with cancer are stable, some authors recommend isolation at home combined with telemonitoring [6,29,31,33,37,38].

### 3.4. Chemotherapy and Other Treatments

There is still controversy as to whether chemotherapy should be delayed in asymptomatic COVID-19 patients due to the current absence of evidence that chemotherapy may result in further harm for these patients [9,38]. There are articles that include more stringent measures regarding the possibility of continuing or initiating specific anticancer treatment [36,37,39,40]. In particular, Wu Xiaoyan et al. (2020) recommend immediately interrupting chemotherapy in the event that a pediatric patient is positive for COVID-19, whether they are with or without symptoms. Instead, others, such as Chao Yang et al. (2020) and Balduzzi et al. (2020), recommend, if the conditions related to the underlying disease are stable, to consider shorter or milder cycles of chemotherapy. Chemotherapy may be delayed for 8–14 days or two negative swab tests should be obtained in the event of asymptomatic cases [28,35,38]. On the other hand, He Yulei et al. (2020), who specifically focused on pediatric patients affected by acute lymphocytic leukemia and acute non-lymphocytic leukemia who already have a chemotherapy program, recommend to not interrupt it during induction treatment and not to delay it for more than 7 days of treatment for the intermediate or the consolidation phase if COVID-19 infection is confirmed.

The same measures were discussed by Hrusak et al. (2020), who conducted a survey in 25 countries examining approximately 10,000 patients who were receiving chemotherapy and immunosuppression. They found that none of the 20 COVID-19-positive children appeared to be severely affected by the infection in association with their treatment. He Yulei et al. (2020) also emphasized that pediatric patients affected by lymphoma or other solid tumors should be treated (after screening for COVID-19) according to their chemotherapy program, and immediately until they fully recover. If a patient is fully recovered, they recommend postponing treatment for no more than 7 days to allow for a period of observation to evaluate their clinical condition due to COVID-19.

Finally, for patients who are recovering and under maintenance chemotherapy, treatment can be delayed for no more than 14 days [9]. It is also suggested that a swab test for COVID-19 be performed before starting any kind of chemotherapy [9,31,35]. In an article that analyzed the experience of a major hospital in North India that receives pediatric patients up to 500km away, chemotherapy was administered at home or in neighboring hospitals by pediatricians, avoiding interruptions and long journeys [27]. Some articles included recommendations on when to administer other types of treatment, such as radiotherapy or surgery. Radiotherapy did not substantially affect the immune system; therefore, its continuation is recommended [26,28,32,35,36,40,45]. On the other hand, for surgery, patients that are suspected to be or are COVID-19-positive have to be transferred to units that are appropriate for pre-operative treatment and then be taken to the operating room through specially dedicated pathways [32,33,35,36,39,40].

In the case of pediatric patients requiring hematopoietic stem cell transplantation (HSCT), postponement is recommended for the majority of non-malignant diseases. HSCT postponement is not recommended for malignant diseases. In addition, all donors and recipients must be tested to confirm the exclusion of COVID-19 infection [28,29,33,37,38].

### 3.5. Remote Support

Various recommendations have been made in relation to remote support. First of all, scholars underline that remote support can be adopted if the patients’ conditions allow for it, and especially if family members or caregivers can support them through this system [6,41]. In the event of an emergency, such as acute symptoms, the patient is transferred or has to go to hospital. In this context, telemedicine and especially video consultations have been promoted and expanded to reduce the risk of contamination. This adoption was shown in three out of the seven articles in order to reduce to the minimum non-urgent visits to the outpatient clinic, such as cancer pediatric patients coming for follow-ups [6,31,33,37,41,43]. Another widely used strategy, which was also easier to use, was telephone consultations. This has to be limited to patients and family members who only need general information about COVID-19 or for pre-triage before coming to hospital [6,27,29,31,33,37,38,41]. Telephone consultations can be combined with an online system to provide further support [31,38,41]. This online system enables the analysis and categorization the information received by the public during pre-triage in order to monitor symptoms; plan interventions to provide specialized support at home (e.g., swab tests, medications, aids); and, consequently, initiate teleconsultation (audio and/or video) and telemonitoring. In the study by Trehan et al. (2020), also in North India, the use of “WhatsApp” groups were allowed between doctors and hematology nurses and patients for assistance, information, and drug prescriptions [27].

In the event that hospital access is necessary, access to the wards is often denied to carers, visitors, and volunteers. Limited access to only one parent is recommended for the visits to the pediatric units and clinical areas to reduce/prevent the spread of COVID-19 in children with cancer [29,33,37,41,43,44]. Both parents are allowed only if specific medical procedures or interviews/consent are required [29,38,43]. Visits have to be reduced also for visitors and/or volunteers [6,29,37,38,43,44].

### 3.6. Dedicated Staff, Pathways, and Continuing Healthcare Education

The establishment of dedicated staff, especially in the transplant units [38] or a panel of COVID-19 experts [9] who make multidisciplinary medical decisions, was another measure identified through our review. Specifically, these dedicated staff should collaborate closely with the COVID-19 medical staff and make decisions jointly [27,30,33,37,38,39,43,44]. In addition, health workers caring for immunocompromised patients should be separated into groups without mutual physical contact. This could be achieved by working every other day (unless the workload does not allow this) or every other week so that they avoid staying in the same offices and common areas [9]. In one study, it was also recommended that backup staff be always available [36]. Finally, staff members who have had contact with positive cases without protective equipment must stay at home in isolation for 14 days [37,39] or test negative on two consecutive swabs [44].

Balduzzi et al. (2020) also underline the importance of identifying better pathways and strategies to prevent the spread of COVID-19 among pediatric patients, medical and nursing staff, and parents involved in their children’s care. These include separate “clean” and “COVID-19” pathways to reach the wards, and the identification of clean areas and those dedicated to COVID-19 patients [9,27,29,32,34,36,37,38,44]. Most of the specialists see their patients in their rooms in order to avoid them from going around the hospital as much as possible [38]. A more detailed article on how to structure the pathways for children with cancer was conducted by Yulei et al. (2020), who identified the need to create four separate zones within each hospital to reduce the incidence of cross-infection and to screen patients potentially infected with COVID-19 [42]. Moreover, Cai et al. (2020) describe a division into three areas: a clean area, a potentially polluted area, and a contaminated area. Each area and aisle must be provided with striking ground signs and space signs that clearly indicate what area it is [39]. In the end, detailed information on how to disinfect environments is provided [39]. Moreover, correct waste disposal systems are highly recommended in five of the included articles [9,26,30,33,37].

Strategies to foster education included weekly educational sessions to update all health workers on the latest information and the latest changes in the guidelines regarding cancer patients affected by COVID-19. In addition, health workers received education regarding COVID-19 symptoms for their daily self-assessment. Another strategy was to provide educational booklets for cancer patients to ensure continued education for both health workers and patients [9,26,30,31,33,37,39]. In addition, scholars suggested the distribution of booklets illustrating the characteristics and risks of COVID-19 infection in children with cancer at the hospital reception, or by e-mail to provide further information to families who needed them [38]. Continued education could also be provided through dedicated websites or systems [31,41].

## 4. Discussion

Considering there are only a few high-quality articles included in this review that are capable of measuring the outcomes described, it is difficult at the moment to draw evidence-based recommendations from their conclusions. Through this rapid review, we identified interventions and strategies to prevent the spread of COVID-19 in children with cancer. This mainly depends on the local COVID-19 situation and other local factors, such as the country’s economic, political, and social aspects. Some of these are common to all and in line with the general recommendations issued by the WHO. Other more specific factors depend on the local epidemiological situation. Finally, another aspect is the development level of the country in which the study was conducted. However, considering the low number of articles from low- and middle-income countries compared to high-income countries, it is very difficult to draw conclusions on the basis of this aspect.

The geographical origin of the articles across five continents shows how the need to protect fragile populations, such as children with cancer, is shared at the international level among high-, low-, and middle-income countries. However, it is important to underline that all the interventions adopted for these patients are in line with the available evidence. The studies included in this review reported benefits from the measures implemented but did not accurately measure the effectiveness of their recommendations. Strategies included limiting the risk of contagion by restricting the access of accompanying persons and parents; banning volunteers and teachers from entering the wards [46,47,48,49]; implementing hygiene measures for patients, health workers and parents, with a particular focus on handwashing; and the use of personal protective equipment (e.g., surgical masks) for staff, patients, health workers, and parents that provide care. Equally important is the identification of separate pathways for the management of patients who were suspected or confirmed to be infected with COVID-19, in agreement with the center of reference for infectious diseases. Just as crucial is the postponement of people accessing the hospital for non-urgent or unnecessary tests or medical examinations, as well as the preventive screening of patients before chemotherapy treatment or transplantation of hematopoietic stem cells, whenever required by the local epidemiological situation or by the directives of the Italian Regional Governments or relevant scientific societies [8,46,50,51,52,53].

Most of the pediatric clinics that care for cancer patients [7] are experiencing or have experienced the adaptation process of management plans aimed at preventing the spread of the new coronavirus [4]. The preventive measures implemented by single organizations should be in line with the official guidelines issued by the relevant scientific societies, which comply with the international directives issued by the WHO and have been adapted according to local needs. These guidelines should be constantly updated according to the evolution of the epidemiological situation, taking into account the specific characteristics of children with cancer. For example, this review reports two expert consensuses referred to different geographical contexts, European [33] and Indian [32]. The European document described most of the preventive interventions reported in this rapid review, whereas the Indian consensus mainly focused on treatment management and other aspect of patient management, such as dedicated pathways and follow-up visits. It is worth noting that the latter interventions were not described in other guidelines [31] or recommendations [35].

With regard to the administration of chemotherapy and other treatments, the present review highlights that the indications are also in line with the recommendations implemented in other contexts. In particular, not interrupting treatments if they cause significant immunodepression and implementing all the preventive measures, starting from structural ones. The postponement of chemotherapy cycles has to be taken into account by groups of experts and be carefully evaluated [38]. Other elements involved checking for people and patients who could be COVID-19-positive and postponing non-urgent surgical operations [8,26,51,54,55,56,57], not to exclude the possibility that children with cancer could be infected with COVID-19. Therefore, it is very important that scientific societies from all over the world issue precise protocols and guidelines to be rigorously followed in these cases.

Finally, it is very difficult to measure which of these interventions are effective in preventing the spread of COVID-19. Excessive interventions can be useful during an emergency when the dynamics related to the way the virus spreads are still not fully known, but urgent considerations and studies are needed to understand which of these interventions are truly effective and useful for these patients and their families. This issue becomes crucial, especially when considering the fact that children affected by cancer and their families need to live without risk or with reduced risk, despite the spread of COVID-19 in the general population.

## 5. Conclusions

Although most of the articles included in this rapid review were descriptive, heterogeneous, and mainly presented experiences with reference to the first pandemic wave, they report several interventions that could be useful to help organizations manage children with cancer during the COVID-19 emergency. While not all contexts are the same, we know there are interventions that could be universally adopted. The purpose of this review was to respond to a common problem [9] and understand if all the measures adopted to prevent the transmission of COVID-19 are actually necessary and supported by evidence. Every preventive and organizational measure to protect the safety of children with cancer needs to be carefully evaluated in order to understand whether to keep it or not. Specifically, special attention should be given to the postponement of treatments that could become an issue for these children’s future prognosis. Instead, there are some restrictive measures such as quarantine, which could have universal consequences, leading to new delayed diagnoses, important psychological problems, and delays in basic care producing significant negative outcomes. In conclusion, we hope that all the preventive measures adopted to limit the spread of COVID-19 may serve as a lesson and guidance for similar pandemics in the future.

## 6. Limitations

The first limitation is the small number of papers included in the present rapid review. Another limitation is the low quality of the studies, which were all descriptive papers that reported experiences, actions, and general outcomes, with no details on how they were measured. The included articles were also very heterogeneous in terms of population, implemented interventions, and outcomes, and this made it difficult to understand if and in what way the single interventions influenced the outcomes. With regard to sample selection, this rapid review only focused on preventive or management support interventions, which may not be representative of all the interventions. Moreover, the local situation of each center may have had an important influence on the type of interventions, but also how these interventions were implemented or can be followed. Finally, this review included only articles conducted in pediatric cancer contexts. In the future, it could be useful to widen the investigation to include other immunosuppressed pediatric patient populations, such as those affected by sickle cell anemia or other immunosuppression diseases.

## Figures and Tables

**Table 1 jcm-09-03756-t001:** Date extraction of the included studies.

First Author, Year	Region/Country	Aim	Study Design or Article Type	Population and Setting	Intervention Area
Subspecialty Group of Hematology and Oncology, 2020	Hubei/China	To develop the management guideline for pediatric wards of hematology and oncology during COVID-19	Practice guideline	• Children with hematological tumors • Parents • Staff ∘ Hematology units	• Management • Education • Prevention • Onco-hematologic treatment management
Yang et al., 2020	South Central/China	To provide suggestions on how to choose a reasonable treatment strategy between epidemic prevention and anticancer therapy under the current epidemic conditions	Letter to the editor	• Children with cancer ∘ Oncology unit	Prevention Oncologic treatment management
Kotecha, 2020	Western Australia/Australia	Summary of recommendations to identify strategies for preventing and treating COVID-19	Correspondence	• Children with cancer • Parents • Carers ∘ Oncology unit	• Management • Education • Prevention • Onco-hematologic treatment management
Bouffet et al., 2020	Greater London/United Kingdom	To facilitate the dissemination of helpful information and useful links, and to evaluate the impact of COVID-19 in the practice of pediatric oncology	Commentary	• Pediatric oncology population • Parents • Carers • Staff ∘ Oncology unit	Management Prevention
Balduzzi et al., 2020	Lombardy/Italy	To share pediatric hemato-oncology issues encountered in the first few weeks of the COVID-19 outbreak in Italy and to alert healthcare professionals worldwide to be prepared accordingly	Review	• Children with hemato-oncology tumor ∘ Hemato-oncology unit ∘ Transplant unit ∘ Outpatient unit	• Management • Prevention • Onco-hematologic treatment management
He et al., 2020	China	To propose a strategic plan for the management of COVID-19 outbreaks in pediatric hematology and oncology departments, focusing primarily on viral infection prevention and control strategies	Comment	• Children with hemato-oncology tumor • Carers • Staff ∘ Pediatric hemato-oncology unit ∘ Outpatient unit	• Management • Education • Prevention
Hrusak et al., 2020	25 different countries (Europe/Asia/Oceania/South America)	• To explore the incidence and severity of COVID-19 among children on anticancer treatment • To describe preventive measures that are in place or should be taken and treatment options in immunocompromised children with COVID-19	Survey	• Children treated with chemotherapy or intensive immunosuppression ∘ Hemato-oncology unit	Prevention Onco-hematologic treatment management
Saab et al., 2020	Middle East/North Africa/West Asia Region	• To investigate the impact of the pandemic and its associated response on the care of children with cancer • A survey, focusing on challenges in pediatric oncology management and barriers to care during the COVID-19 pandemic	Survey	• Pediatric oncology population ∘ Oncology unit	• Management • Prevention • Oncologic treatment management
Cassoux, 2020	Paris/France	To provide recommendations on ocular cancer management during COVID-19	Recommendations	• Pediatric ocular cancer population ∘ Department of Ophthalmology	Management Prevention
Manjandavida et al., 2020	India	To establish guidelines and recommendations for ocular oncology in the management of ocular tumors Medical pediatric oncology guidelines	Expert consensus	• Pediatric ocular oncology ∘ Oculoplastic service	• Management • Oncologic treatment management
Trehan et al., 2020	North India	How were pediatric oncology patients managed in a lower middle-income country during the COVID-19 pandemic	Letter to the editor	• Children with hemato-oncology tumor • Staff ∘ Pediatric hemato-oncology unit ∘ Outpatient unit	Management Prevention
Cai et al., 2020	Zhejiang/China	Prevention and control strategies in the diagnosis and treatment of solid tumors in children during the COVID-19 pandemic	Review	• Pediatric oncology population • Staff ∘ Oncology department	Education Oncologic treatment management
Iehara et al., 2020	Japan	Prevention and treatment of COVID-19 in pediatric cancer patients in Japan	Letter to the editor	• Children with hemato-oncology tumor ∘ Hemato-oncology unit ∘ Transplant unit	Management Prevention Onco-hematologic treatment management
Sullivan et al., 2020	Europe	To provide some practical advice for adapting diagnostic and treatment protocols for children with cancer during the pandemic, the measures taken to contain it (e.g., extreme social distancing), and how to prepare for the anticipated recovery period	International clinical consensus	• Children with cancer • Parents • Carers • Staff ∘ Pediatric hemato-oncology unit ∘ Transplant unit ∘ Outpatient unit	Management Prevention Education Onco-hematologic treatment management
Amicucci, 2020b	Italy	To describe a preventive and control measures to manage COVID-19 infection in Italian Pediatric Oncology and Hematology Association (AIEOP) centers	Letter to the editor	• Children with hemato-oncology tumor • Parents • Carers ∘ Hemato-oncology unit ∘ Transplant unit ∘ Outpatient unit	Management Prevention Onco-hematologic treatment management
Baruchel et al., 2020	France	To provide specific recommendations for preventing and for treatment management to care for children and adolescents with acute lymphoblastic leukemia during the COVID-19 pandemic	Recommendations	• Children with hematology tumor ∘ Hematology unit	Management Prevention Hematologic treatment management
Ruggiero et al., 2020	Rome/Italy	To manage the COVID-19 outbreak in children with cancer	Editorial	• Children with cancer • Parents • Carers • Staff ∘ Oncology unit	Management Prevention
Seth, 2020	New Delhi/India	Project the challenges faced by children undergoing treatment and the treating pediatric oncologist during the COVID-19 pandemic	Correspondence	• Children with cancer • Staff ∘ Oncology unit	Management Prevention
Amicucci, 2020a	Italy	To explore COVID-19 containment measures adopted by Italian Pediatric Oncology and Hematology Association (AIEOP) centers to prevent the virus spread among healthcare providers	Letter to the editor	• Children with hemato-oncology tumor • Staff ∘ Hemato-oncology unit ∘ Outpatient unit ∘ Home care	Management Prevention Education
Kaspers, 2020	Amsterdam/the Netherlands	Prevention and impact of COVID-19 in children with cancer	Editorial	• Children with cancer • Parents • Carers • Staff ∘ Oncology unit	Management Prevention
Sainati and Biffi, 2020	Padova/Italy	Describe how it was to deal with the COVID-19 epidemic in an Italian pediatric onco-hematology clinic located in a region with a high density of cases	Correspondence	• Children with hemato-oncology tumor • Parents • Carers ∘ Hemato-oncology unit ∘ Transplant unit ∘ Outpatient unit ∘ Home care	Management Prevention Education Onco-hematologic treatment management

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
