# Peer review of "The Management of Children with Cancer during the COVID-19 Pandemic: A Rapid Review"

_jcm, 2020, doi:10.3390/jcm9113756_

Round 1

Reviewer 1 Report

This is a revised version of a paper dealing with a systematic rapid review to analyse the scientific literature regarding the management, interventions and strategies to prevent the spread of COVID-19 in the paediatric cancer population.

The commenst of the reviwer were taken seriously and sufficiently answered.

Minor point:

Line 105-106: the sentence: 'In all the A hospital unit was the setting for every included article.' The meaning of the sentence is unclear. What is a A hospital unit? This needs to be made understandable.

Author Response

Please see the attachment: "Reviewer_1_JCM"

Reviewer 2 Report

reviewer criticisms have been addressed and recommend publication 

Author Response

Please see the attachment: "Reviewer_2_JCM"

This manuscript is a resubmission of an earlier submission. The following is a list of the peer review reports and author responses from that submission.

Round 1

Reviewer 1 Report

The authors present a rapid review about the management of children with cancer during the COVID-19 Pandemic. Out of 505 identified articles only 21 could be included in their analysis.

The follwoing results of the review are most important and clearly elaborated:

  1. Quality of most papers is poor, very heterogeneous, only descriptive and presenting mainly experiences. In their seection are also Editorials and a Letter to the Editor.
  2. It can not be concluded which interventions influences outcome.

As the interventions are also depending on the local COVID-19 situation and other local factors, it would be of interest, if the interventions in low and middle income countries are different from those in high income countries. This may be difficult to answer, as the number of papers is small. Nevertheless the local situation may have a great influence on the kind of interventions, but also how these interventions are followed or can be followed. This may be another limitation that should be discussed.

Another issue that may be of interest is the fact that there is one paper from Sullivan et al, which is an Ingternational clinical consensus paper and a second one from India as an expert consensus paper (Manjandavida et al.) to compare the recommendations of these two papers, if they are similar and to look how many of the other papers are in line with the experts. It would be interesting to ellaborate on this topic.

Author Response

Thank you for the time devoted to our manuscript.

Reviewer 2 Report

In the manuscript "The Management of Children with Cancer During the COVID-19 Pandemic: A Rapid Review" the authors discuss recent papers in regards to management of pediatric patients.  While this could be a result of the state of the pandemic, there are very few conclusions being drawn from this manuscript that will influence management. The manuscript does a decent job of filtering recent manuscripts, but a better summary of management across children would have been preferred (but again difficult to to ongoing stage of pandemic).  However, there are subsets of children more greatly affected by covid-19 such as patients with sickle cell anemia, which are not thoroughly discussed.  This is a concerning omission.     

Author Response

(The authors gave the same response as above.)

Round 2

Reviewer 2 Report

Although manuscript is descriptive summary is succinct and well written.